# Active and passive touch are differentially represented in the mouse somatosensory thalamus

Anton Sumser[1]*, Emilio Ulises Isaías-Camacho[2], Rebecca Audrey Mease[2], Alexander Groh[2]*

1 Division of Neuroscience, Faculty of Biology, LMU Munich, Martinsried, Germany, 2 Medical Biophysics, Institute for Physiology and Pathophysiology, Heidelberg University, Heidelberg, Germany

* sumser@bio.lmu.de (AS); groh@uni-heidelberg.de (AG)

## Abstract

Active and passive sensing strategies are integral to an animal's behavioral repertoire. Nevertheless, there is a lack of information regarding the neuronal circuitry that underpins these strategies, particularly at the thalamus level. We evaluated how active versus passive whisker deflections are represented in single neurons of the ventral posteromedial thalamus (VPM) and the posterior medial thalamus (POm) in awake mice. These are the first- and higher-order thalamic nuclei of the whisker system, respectively. VPM neurons robustly responded to both active and passive whisker deflections, while POm neurons showed a preference for passive deflections and responded poorly to active touches. This response disparity could not be explained by stimulus kinematics and only in part by the animal's voluntary whisking state. In contrast, cortical activity significantly influenced POm's responses to passive touch. Inhibition of the barrel cortex strongly attenuated whisker responses in POm and simultaneously increased the whisking phase coding. This suggests that POm receives touch information from the cortex which strongly adapts and is gated by rare events. Together, these findings suggest two thalamic relay streams, where VPM robustly relays both active and passive deflection, while POm's sensitivity requires top-down cortical involvement to signal salient events such as unexpected deflections, originating in the environment.

## Introduction

Animals can employ different sensing strategies to accomplish a given task efficiently. For instance, mice and rats can solve whisker discrimination tasks with or without active whisker movements, by either actively touching an object through whisking (active touch) or by letting the object touch their stationary whiskers (passive touch) [1,2]. While both active and passive sensing strategies are part of a larger behavioral repertoire that provides mammals with adaptive flexibility for solving tasks, there is limited information regarding the neuronal circuitry underlying active and passive strategies, particularly at the level of the thalamus—the obligatory gateway for somatosensory signals en route to the cortex.

Whisker signals are processed through two main thalamic pathways: the ventral posteromedial nucleus (VPM) processes a direct sensory stream from the periphery, while

**Data availability statement:** All the data are in the manuscript or in Supporting information. Source data and code can be found here: https://doi.org/10.5281/zenodo.14691035

**Funding:** This work was supported by the German Research Foundation (https://www.dfg.de/de) (DFG Grant GR3757/4-1 to AG), Boehringer Ingelheim Fonds fellowship (https://www.bifonds.de/fellowships-grants/phd-fellowships.html) (salary grant to A.S.), the Heidelberg Graduate Academy completion grant through the Landesgraduiertenförderung program with funds allocated by the German Ministry of Science, Research and Arts (https://mwk.baden-wuerttemberg.de/en/home) (salary grant to EIC). We acknowledge the data storage service SDS@hd and high-performance computing initiative bwHPC, supported by the Ministry of Science, Research and the Arts Baden-Württemberg (SDS@hd and bwHPC) and the German Research Foundation (DFG) through grants INST 35/1597-1 FUGG (bwHPC) and INST 35/1503-1 FUGG (SDS@hd). The funders had no role in study design, data collection and analysis, decision to publish, or preparation of the manuscript.

**Competing interests:** The authors have declared that no competing interests exist.

**Abbreviations:** AP, action potential; BC, barrel cortex; L5B, layer 5B; MI, modulation index; PBS, phosphate-buffered saline; POm, posterior medial nucleus; PSTHs, peri-stimulus time histograms; VGAT, vesicular GABA transporter; VPM, ventral posteromedial nucleus; ZI, zona incerta.

the posterior medial nucleus (POm) is additionally controlled by a strong indirect sensory stream via the cortex [3–7]. This anatomical segregation of pathways suggests that distinct motor-sensory-motor loops may implement different sensory-motor processes—such as whisker motion, active touch, and passive touch [8]. Previous work supports this idea in that VPM neurons exhibit strong and tightly time-locked spiking responses to passive stimuli applied to stationary whiskers [9]. In contrast, POm neurons respond to a lesser extent and with lower temporal precision. POm's comparably low sensitivity to whisker deflections has been proposed to result from strong inhibition by the zona incerta (ZI) [10]; ZI inhibition in turn was suggested to be controlled by motor cortical output [11]. This motor gating mechanism predicts that sensory transmission through POm is contingent on whisking (motor activity), and thus POm should be sensitive to active touch but not to passive deflections. However, this prediction has never been tested.

Notably, to date little is known about the encoding of active and passive touch signals conveyed by these pathways, in particular via the higher-order thalamus, which subserves diverse cognitive processes (reviewed in [12,13]). This is largely due to the difficulty of recording from identified thalamic neurons aligned to a single whisker touching an object. Therefore, previous studies have either examined the thalamic representations of whisking and passive touch in awake animals [14] or artificially evoked touches in anesthetized animals [8]. Consequently, the lack of a direct comparison between active and passive touch representations hinders our understanding of the specific functions of thalamic nuclei during ecologically realistic touch scenarios in which active and passive touches intermix, for example, during prey hunting [15] or social touch [16].

To approach this problem, we leveraged the ability of juxtacellular recordings to identify individual VPM and POm neurons and scan for those with robust responses to the passive deflection of a single whisker in awake, head-fixed mice. Letting the animals actively touch a metal pole with the same whisker allowed us to directly compare the representation of active versus passive whisker deflections in the same recorded neurons. While passive whisker deflections evoked robust responses in POm and VPM, POm was surprisingly insensitive to active touches. This discrepancy was not due to behavioral state-dependent gating, or kinematic differences between the stimulus types. However, POm neurons showed significant sensitivity to stimulus interval, suggesting that touch information could be conveyed from the barrel cortex (BC) to POm via strongly depressing cortico-thalamic synapses. Indeed, inhibition of the BC strongly attenuated whisker responses in POm, while increasing whisking phase representation, consistent with POm receiving cortical touch information and brainstem phase information. Taken together, these findings suggest a specific, barrel-cortex-supported sensitivity of POm to passive deflections which might signal unexpected events.

## Results

### Thalamic responses to active and passive whisker deflections

We investigated thalamic responses to active and passive whisker deflections by juxtasomally recording single neurons in either VPM or POm in awake and spontaneously whisking head-fixed mice (Fig 1A and 1B). To ensure that only one whisker was stimulated and to measure precise deflection times, all but one (C1 or C2) whiskers were cut (see Methods for the mapping procedure of the aligned whisker). Passive whisker deflections were induced by air-puffs directed at the spared whisker at pseudorandom intervals (median stimulation rate 1.5 Hz; see Methods). Active touches were spontaneously self-generated by the mouse palpating a vertical pole with its spared whisker. The locations of recorded neurons were determined by stereotaxic coordinates and post-hoc localization of labeled neurons (Fig 1C and 1D; Methods).

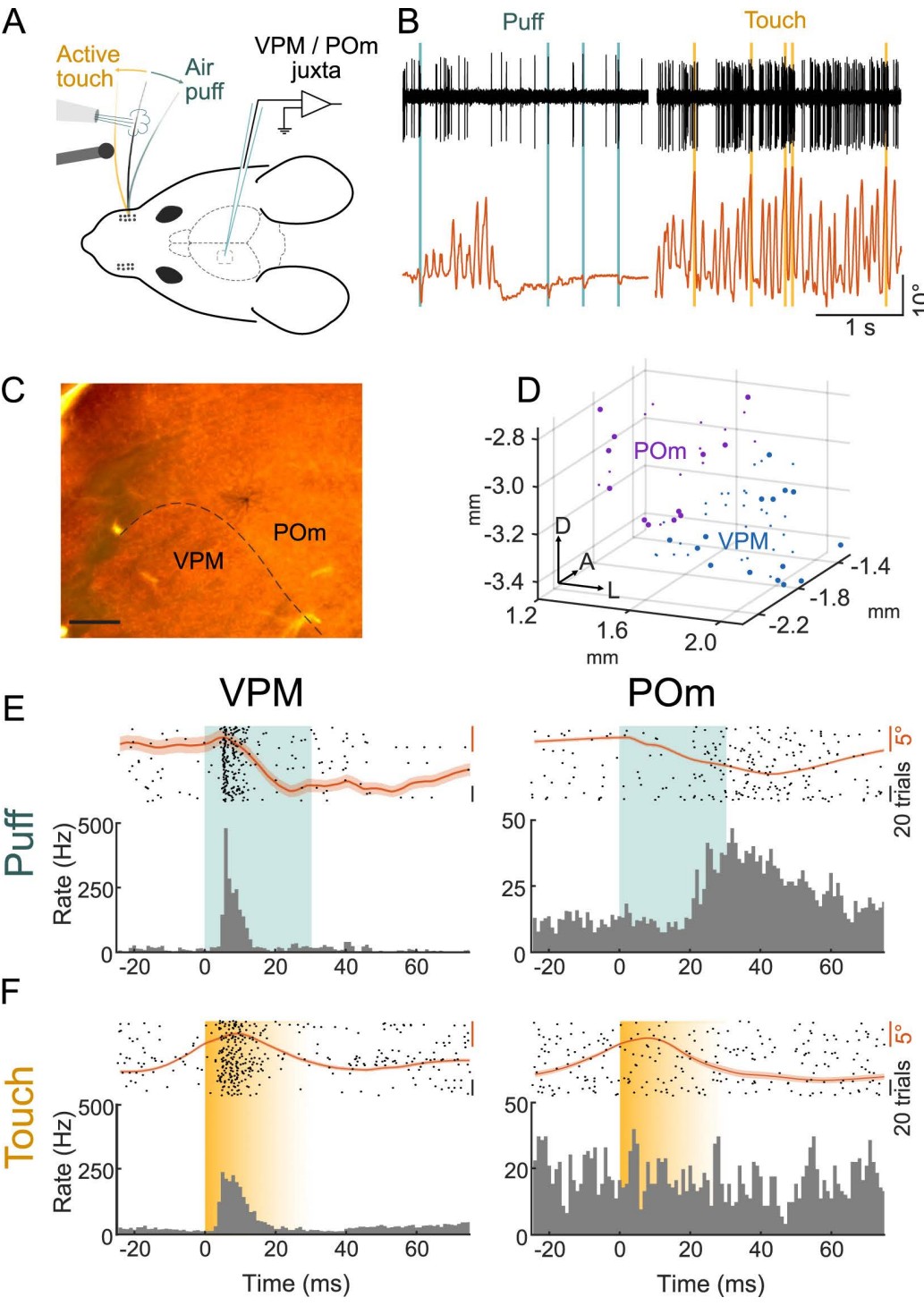

**Fig 1. Location-recovered recordings of single VPM and POm neurons during active and passive whisker deflections in behaving mice. (A)** Experimental paradigm: single whisker deflection by focal air puffs (teal) or by active touches of a pole (yellow). Neuronal responses in VPM or POm are recorded juxtasomally. **(B)** Example band-pass filtered juxtasomal recording of a VPM neuron (black, same neuron as in E) and whisker angle (orange) during air puffs (teal) and active touches (yellow). **(C)** Example labeled neuron in POm. Scale bar = 200 μm. **(D)** Reconstructed locations of recorded VPM (blue) and POm (purple) neurons; large dots indicate neurons with significant air puff responses ($n$ = 16 animals, VPM = 24/43, POm = 14/26, responsive/total recorded neurons; comparison of 50 ms windows before and after air puff, $p < 0.05$ one-sided Wilcoxon signed-rank test). **(E)** Raster plots (top, 50 trials) and PSTHs (bottom)

of representative single neuron responses to puff in VPM (left) and POm (right). Mean whisker angle in orange, SEM indicated as shaded area. **(F)** Same as E but for active touch (yellow) responses. Shading indicates variable touch offset. Data and code underlying this figure can be found here: https://doi.org/10.5281/zenodo.14691035.

We compared POm and VPM responses to active and passive deflections by analyzing spike times relative to whisker deflection onsets (Fig 1E and 1F). To ensure somatotopic alignment between the recorded neurons and the stimulated whisker, only neurons with significant responses to air puff deflections of the spared whisker were included in further analyses (Fig 1D, large dots), the remainder showed neither puff nor touch responses (S1 Fig). For direct comparisons of air puff and active touch responses, we first focused on the subset of those recordings, which contained at least 10 spontaneously generated active touch events ($n$ = 15 VPM neurons in $n$ = 11 animals; $n$ = 11 POm neurons in $n$ = 6 animals).

Within this set of puff-responsive neurons, VPM responded faster than POm neurons (first spike latencies VPM: 15 ± 2 ms versus POm: 27 ± 2 ms, $p$ = 0.003 VPM versus POm, two-sided Wilcoxon rank sum test) and with higher temporal precision (standard deviation of first spike latencies, VPM: 10 ± 2 ms versus POm: 15 ± 1, $p$ = 0.015 VPM versus POm, two-sided Wilcoxon rank sum test) to air puffs (Fig 2A). In contrast, VPM and POm responses to active touches differed substantially (Fig 2B): While most puff-responsive VPM neurons responded to active touches (9/15, 60%), comparatively few puff-responsive POm neurons responded to touch (4/11, 36%).

This response disparity was even more pronounced when comparing passive and active whisker response magnitudes (firing rates at baseline versus response). VPM's response magnitudes were significantly higher than baseline in both passive and active deflection trials (puff: 42 ± 10 Hz; touch: 37 ± 7 Hz, Fig 2C). In contrast, POm's response magnitudes were only significantly higher than baseline in responses to puff but not in response to touch (puff: 21 ± 5 Hz; touch: 17 ± 2 Hz, Fig 2D). As baseline firing differed across active and passive conditions in both VPM and POm (S1 Table) we computed a modulation index (MI) of response firing rates relative to baseline (Fig 2E; Methods). VPM and POm firing rates were more modulated by air puffs (VPM: MI = 0.56 ± 0.06; POm: MI = 0.31 ± 0.05) compared to active touches (VPM: MI = 0.25 ± 0.06; POm: MI = 0.02 ± 0.06). Consistent with a low sensitivity of POm to active touch, POm's firing rate modulation was on the population level not significantly different from zero ($p$ = 0.97, two-sided Wilcoxon signed-rank test).

These response differences were sufficient to classify neurons into VPM and POm neurons solely based on their response profiles using a logistic regression model (see Methods). This functional classification accurately matched our histological classification of location-recovered neurons (100% accuracy, S2 Fig).

## Whisking reduces sensory responses in VPM and POm

Both VPM and POm neurons showed higher baseline activity preceding touches compared to baseline activity preceding air puffs (Fig 2C and 2D). While active touches are always associated with whisking (W), air puffs can occur in both whisking (W) and quiescent (Q) periods. Therefore, one could argue that whisking may suppress or mask touch responses in POm and thereby explain the pronounced response disparity between puff and touch in POm. To disentangle the role of whisking in thalamic responses to whisker deflections, we analyzed whisker movements in more detail for all neurons with air puff responses ($n_{VPM}$ = 24 out of $n$ = 13 animals; $n_{POm}$ = 14 out of 7 animals). Confirming earlier studies [14,17,18] baseline activity in both VPM and POm was roughly twice as high during free whisking compared to quiescence

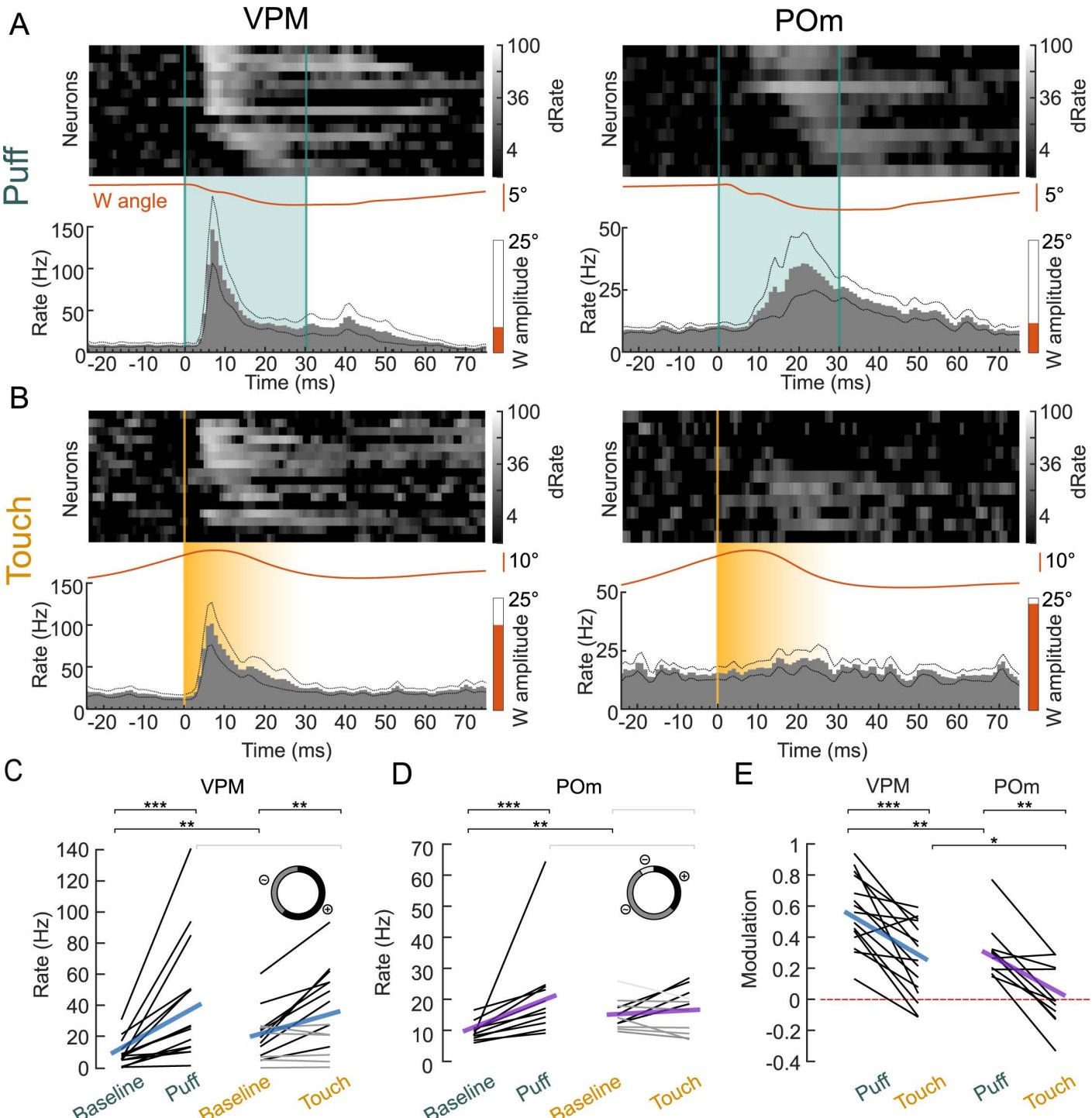

**Fig 2. POm shows pronounced response disparity to active and passive whisker stimuli. (A)** VPM (left) and POm (right) responses to air puffs ("Puff", teal). Top: heatmaps of individual neuron PSTHs, normalized to baseline before puff and ordered by peak response latency. Bottom: mean population PSTHs, mean whisker angles (orange lines), and mean whisking amplitude over the plotted time range (−25 to 75 ms, orange bars) from recordings shown in the top panel. **(B)** Same as A but for active touches ("Touch", yellow). **(C)** VPM spike rates during baseline and after whisker deflection via Puff or Touch. Baseline: 50 ms windows before, Puff/Touch: 50 ms after deflection. Individual neurons (black: rate increase $p < 0.05$ (9/15 neurons), dark gray: non-significant (6/15 neurons)) and population means (blue). Inset: pie chart of neurons with positive, negative, and non-significant responses to active touches. **(D)** Same as C but for POm (purple). Rate increase: 4/11 neurons, rate decrease: 1/11 neurons (light gray), non-significant: 6/11 neurons. **(E)** Modulation of Puff and Touch spike rates in comparison to baseline. Individual neurons (black) and population means for VPM (blue) and POm (purple), zero modulation indicated by red-dashed line. Asterisks represent

*p*-values (\**p* < 0.05, \*\**p* < 0.01, \*\*\**p* < 0.001); 2C–2E between conditions: two-sided Wilcoxon signed-rank test; 2E between neuron populations: two-sided Wilcoxon ranked-sum test; Individual neuron comparison (2C, 2D): one-sided (Puff) or two-sided (Touch) Wilcoxon signed-rank test; exact *p*-values, *N* numbers in S1 Table. Data and code underlying this figure can be found here: https://doi.org/10.5281/zenodo.14691035. Source data for panels C, D, E in S1 Data.

(Rate$_{Q, VPM}$ = 6 ± 1, Rate$_{W, VPM}$ = 13 ± 2 Hz; Rate$_{Q, POm}$ = 6 ± 1, Rate$_{W, POm}$ = 10 ± 1 Hz, Fig 3A). Moreover, we observed that spiking rates increased proportionally with free whisking amplitude in both VPM (mean correlation = 0.69) and POm (mean correlation = 0.81, Fig 3B). Spike timing was also locked to the whisking cycle in about half of the neurons (VPM 14/24 neurons (9/15 VPM neurons with touch recording); POm: 6/14 neurons (5/11 POm neurons with touch recording)), but to a lesser extent as reported in rats [14] (Fig 3C and 3D).

To probe the impact of the whisking state on passive deflection responses, we split the air puff trials into those when the mouse was spontaneously whisking (amplitude > 3°) for at least 500 ms before the air puff (W-Puffs) and those when the mouse was quiescent (Q-Puffs; amplitude < 3°; Fig 3E and 3F). To match the responses to the population in Fig 2, we restricted the analysis to the same population that had active touches recorded ($n_{VPM}$ = 15 neurons in *n* = 11 animals; $n_{POm}$ = 11 in *n* = 6 animals). In both VPM and POm, the response profile was comparable between Q-Puffs and W-Puffs and most (VPM: 11/15, POm: 7/11) neurons responded significantly to W-Puffs despite elevated baseline rates during W-Puffs (Fig 3G and 3H). On the population level, modulation of firing rates upon deflection (Fig 3I) were significantly smaller in both VPM and POm during W-Puff trials (MI$_{VPM, W-Puff}$ = 0.46 ± 0.06, MI$_{POm, W-Puff}$ = 0.22 ± 0.07) compared to Q-Puff trials (MI$_{VPM, Q-Puff}$ = 0.67 ± 0.06, MI$_{POm, Q-Puff}$ = 0.38 ± 0.06; p$_{VPM, Q vs W}$ = 6.1E-05, p$_{POm, Q vs W}$ = 0.002, two-sided Wilcoxon signed-rank test). This attenuation of whisker responses in VPM and POm during whisking is in agreement with earlier observations of top-down sensory gain control accounting already at the brainstem level (Chakrabarti and Schwarz 2018). Population touch firing rate modulation was however still smaller than W-Puff modulation (Fig 2J) in both VPM and POm (p$_{VPM, W vs T}$ = 0.012, p$_{POm, W vs T}$ = 0.024, two-sided Wilcoxon signed-rank test).

Thus, while free whisking elevates thalamic firing rates, and concomitantly reduces relative air puff responses in VPM, differences in whisking state can only partly account for the POm response disparity between air puffs and active touches (Fig 2). Furthermore, this result suggests that whisking does not gate sensory transmission through POm, in contrast to previous reports [11].

## Dependence of deflection responses on kinematic variability and sensory adaptation

An important consideration for the observed disparity between active and passive touch responses in mouse somatosensory thalamus is whether these neuronal response differences originate from kinematic differences between the puff and touch stimuli. To address this possibility, we asked, firstly, if and how kinematic stimulus parameters are encoded in VPM/POm and secondly if different kinematics of air puffs and active touches can explain response disparities. For example, if POm were sensitive to impact strength (i.e., responding more with stronger acceleration or curvature of the whisker), one might conclude that touches are not represented in POm, because touch impact strength might be actively minimized by the animal [19], in contrast to puffs, which are not under control of the animal. Therefore, we analyzed whisker kinematics around each individual deflection and compared neuronal responses to active versus passive deflections within trial groups with matched respective kinematic parameters.

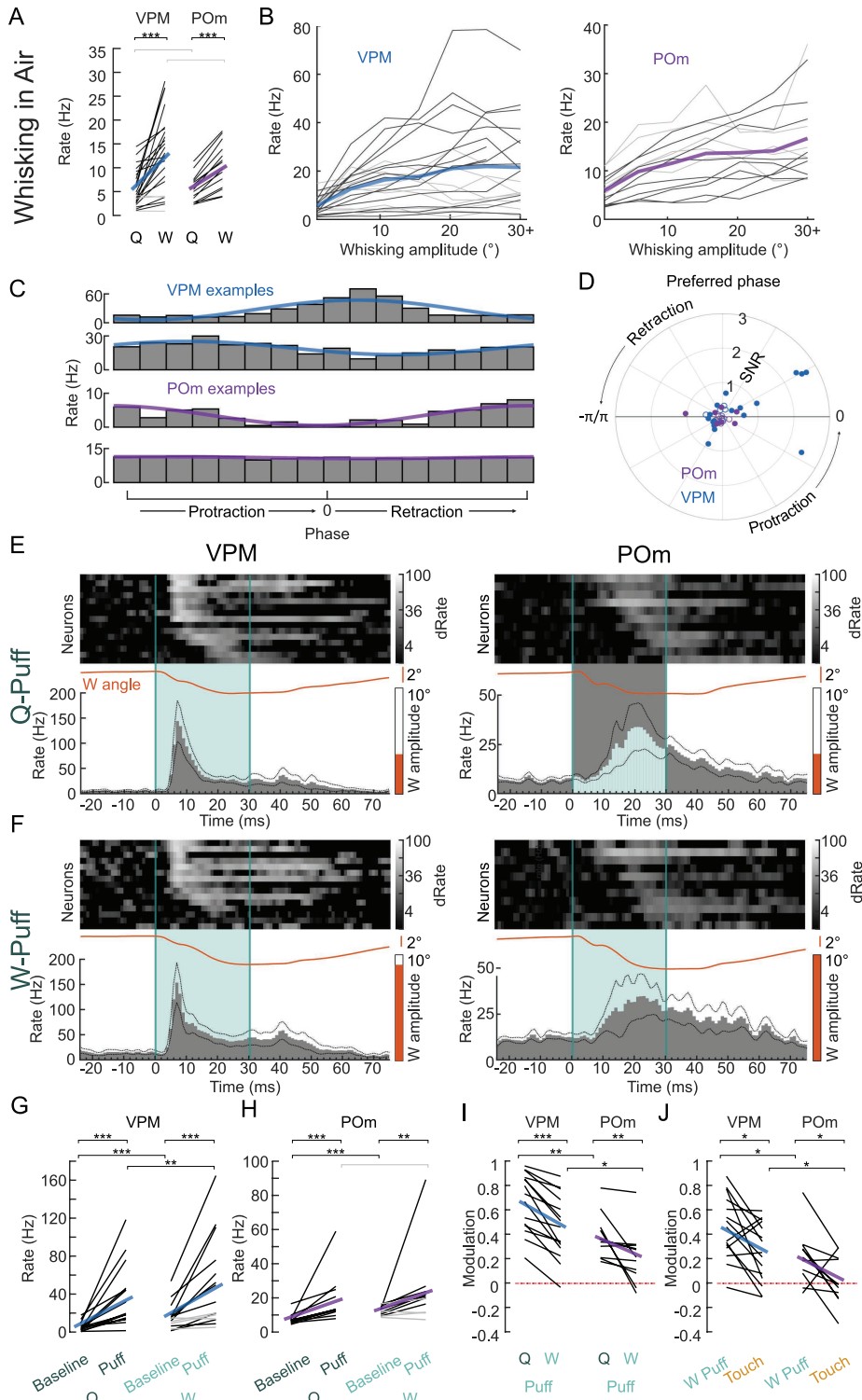

**Fig 3. Influence of whisking on whisker deflection responses in VPM and POm.** (**A**) Spike rates during quiescence (Q = no whisking) and during whisking (W) for individual neurons (black: whisking rate significantly different from base rate ($p < 0.05$, shuffle test, see Methods), gray: non-significant change) and population means (VPM: blue line, POm: purple line). (**B**) Spike rates as a function of whisking amplitudes for VPM (left) and POm (right), individual neurons (black: significant correlation $p < 0.05$, gray: non-significant increase, based on Student $T$ cumulative distribution function), averages in blue and purple for VPM and POm, respectively. (**C**) Spike rates as a function of phase

in the whisk cycle (gray bars) and a sinusoidal fit to the data (blue and purple for VPM and POm example neurons, respectively). **(D)** Polar plot of phase signal to noise (SNR, radius) and preferred phase (angle) for VPM (blue) and POm (purple) neurons. Filled circles, neurons with significant phase modulation ($p < 0.05$, Kuiper test). Phase zero refers to a fully protracted position. **(E)** Responses to air puffs (teal) during quiescence (Q-Puff) in VPM (left) and POm (right), PSTHs of individual neurons (heat maps, normalized to baseline before air puff and ordered by peak response latency) and population PSTHs. Mean whisker angles (orange lines) and mean whisking amplitudes (orange bars). **(F)** Same as E but for air puffs during whisking (W-Puff). **(G)** VPM spike rates during baseline and in response to air puffs when the animal was quiescent (Q) or whisking (W) Baseline: 50 ms windows before, Puff: 50 ms after. Individual neurons (black: rate increase $p < 0.05$, gray: non-significant increase) and population means (blue). **(H)** same as G but for POm (purple) spike rates in comparison to baseline. **(I)** Modulation index of Q-Puff and W-Puff spike rates in comparison to baseline. Individual neurons (black) and population means for VPM (blue) and POm (purple), zero modulation indicated by red-dashed line. **(J)** Modulation index of W-Puff and Touch spike rates in comparison to baseline. Individual neurons (black) and population means for VPM (blue) and POm (purple), zero modulation indicated by red-dashed line. Asterisks represent $p$-values (*$p < 0.05$, **$p < 0.01$, ***$p < 0.001$); 3G–3J between conditions: two-sided Wilcoxon signed-rank test; 3I–3J between neuron populations: two-sided Wilcoxon ranked-sum test; Individual neuron comparison (3A, 3G, 3H): one-sided Wilcoxon signed-rank test; exact $p$-values, N numbers in S1 Table. Data and code underlying this figure can be found here: https://doi.org/10.5281/zenodo.14691035. Source data for panels A, G, H, I, J in S2 Data.

To this end, we split the respective joint active touch and puff kinematic distributions into tertiles (Figs 4A, 4D, 4G, and S3) and analyzed pooled neuronal firing rate modulations in each tertile separately. We first looked into absolute angular acceleration at the base of the whisker as a proxy for impact strength (Fig 4A). VPM neurons ($n_{VPM}$ = 15 neurons in $n$ = 11 animals) showed significant firing rate modulation in all acceleration tertiles in response to air puffs as well as in response to high-acceleration (3rd tertile) active touches (Fig 4B and 4C and S2 Table). POm neurons ($n_{POm}$ = 11 in $n$ = 6 animals) responded more strongly to low-acceleration compared to high-acceleration air puffs (1st versus 3rd tertile $p = 0.019$, Two-sided Wilcoxon signed-rank test), yet displayed no significant modulation in response to active touches, irrespective of acceleration (Fig 4B and 4C). Notably, the distribution of touch acceleration values entirely encompassed the puff distribution (Fig 4A), thus the same acceleration values lead to POm responses in case of passive deflections but not in case of active deflections. Splitting deflection trials into matched normalized curvature tertiles just after the deflection (50 ms), as a proxy for the whisker bending force (Fig 4D) [20], we found significant VPM responses in all active and passive stimulus conditions (Fig 4E and 4F). In contrast, POm neurons showed responses only upon lower curvature puff deflections but not in any of the matched active touch conditions (Fig 4E and 4F). Splitting deflections into tertiles depending on other kinematic parameters like curvature change, setpoint, or amplitude (S3 Fig and S2 Table), similarly led to slight variations in VPM and POm responsiveness, whereas POm neurons selectively only responded to puff deflections but not active touches. Additionally, POm but not VPM neurons were less responsive to air puffs occuring in far protracted whisker positions or during the protraction phase (S3H and S3I Fig), suggesting a contribution of the whisker position to active and passive deflection discrimination. In summary, both POm and VPM neurons were sensitive to puff stimuli, even if kinematic effects on the whisker were small. VPM responses to active touches were in general slightly weaker (Figs 2E, 4B, and 4E) than to air puffs, while POm neurons did not show significant responses to active touches, irrespective of the kinematic features tested.

The only deflection parameter with a substantial influence on puff versus touch responses in POm was the inter-deflection interval. As mice tend to whisk in bouts, intervals between active touches were on average relatively short (active touch interval = 98 ± 149 ms); air puff intervals encompassed the range of short active touch intervals but were on average longer (652 ± 846 ms; median ± interquartile range), Fig 4G. Splitting the joint interval distribution

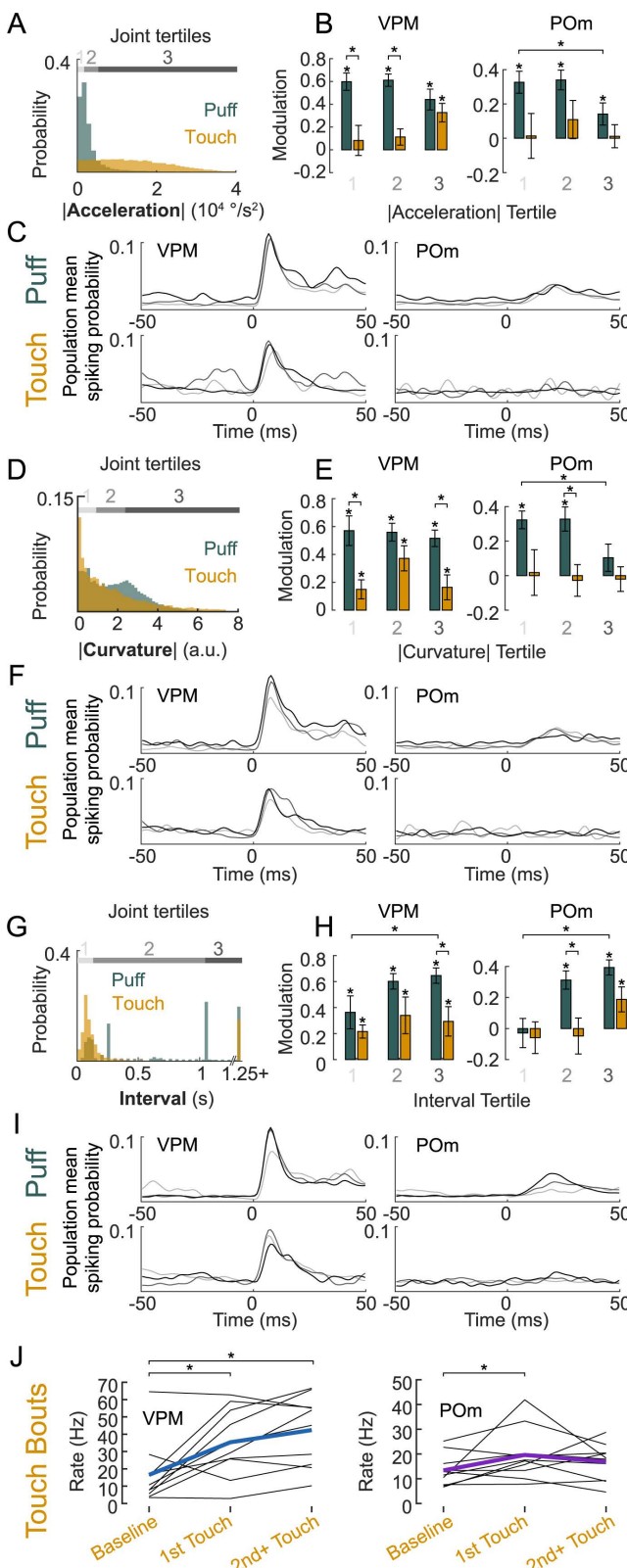

**Fig 4. Dependence of thalamic responses to active and passive whisker deflections on whisker kinematics and intervals.** (**A**) Distribution of absolute angular acceleration during (±25 ms) touch (yellow) and puff (teal). Joint distribution divided into acceleration tertiles (gray-shaded bars). (**B**) Modulation in response to puff (teal) and touch (yellow) split by acceleration tertiles indicated in A for VPM (left) and POm (right). (**C**) Smoothed VPM (left) and

POm (right) population mean PSTHs split by tertiles (shades of gray) indicated in A for puff (top) and touch stimuli (bottom). **(D–F)** Same as in A–C but splitting deflection events into tertiles of whisker curvature following whisker deflection (0–25 ms). **(G–I)** Same as in A–C but splitting deflection events into tertiles based on intervals between whisker deflections. **(J)** Responses to 1st vs. subsequent touches within a bout. Individual neurons (black) and population means for VPM (blue, left, $n = 10$) and POm (purple, right, $n = 10$). Baseline: 50 ms windows before 1st touch in bout, Touch: 50 ms after. Only neurons with at least 10 touches in each condition were included. Asterisks represent significance (*$p < 0.05$); 4B, 4E, 4H within condition: one-sided Wilcoxon signed-rank, between tertiles and conditions: two-sided Wilcoxon signed-rank test; exact $p$-values in S2 Table. Data and code underlying this figure can be found here: https://doi.org/10.5281/zenodo.14691035. Source data for panels B, E, H, J in S3 Data.

into tertiles revealed that VPM neurons responded significantly to all interval tertiles, however with a preference for longer intervals in the case of air puffs ($MI_{Puff, Interval 1} = 0.36 \pm 0.13$, $MI_{Puff, Interval 3} = 0.65$, $p_{Puff, 1 vs. 3} = 0.049$, two-sided Wilcoxon signed-rank test, Fig 4H and 4I and S2 Table). POm neurons in contrast were highly sensitive to inter-deflection intervals: POm responses could not be evoked by short inter-puff and inter-touch intervals ($MI_{Puff, Interval 1} = -0.03 \pm 0.01$, $MI_{Touch, Interval 1} = -0.06 \pm 0.1$). In contrast, long inter-puff and inter-touch intervals evoked significant responses in POm ($MI_{Puff, Interval 3} = 0.39 \pm 0.05$, $MI_{Touch, Interval 3} = 0.19 \pm 0.08$, Fig 4H and 4I and S2 Table). Thus, sensory adaptation during whisking bouts may contribute to the low touch sensitivity in POm. Indeed, POm neurons on average responded only to the 1st touch within a touch bout, while VPM neurons responded to subsequent touches as well (Fig 4J, touch bouts separated by at least 500 ms).

In summary, VPM neurons exhibit sensitivity to deflection kinematics, while POm neurons are sensitive to deflection timing. Notably, substantial responses to active touch deflections in POm exclusively occur with long inter-touch intervals of 1 second or more and only the least adapted response to the first touch in a bout is significantly larger than baseline.

## Cortical dependence of thalamic responses

POm's sensitivity to long inter-deflection intervals (Fig 4G–4I), suggests that POm's whisker responses are primarily conveyed via corticothalamic layer 5 synapses, which are known to be strongly depressing [21,22]. To directly test corticothalamic contributions to POm's responsiveness in behaving animals, we silenced the BC via optogenetic stimulation of Channelrhodopsin-2-expressing inhibitory vesicular GABA transporter (VGAT) neurons (Fig 5A). In accord with [17], VPM's ($n_{VPM} = 9$, $n = 5$ animals) spontaneous activity during whisking and quiescent periods was largely unaffected by cortical silencing, with an approximate 2-fold rate increase when whisking (Fig 5B and S1 Table). In contrast, POm ($n_{POm} = 11$, $n = 4$ animals) activity was strongly attenuated by cortical silencing in both quiescence ($Rate_{Q, Laser Off} = 5.8 \pm 0.9$ Hz versus $Rate_{Q, Laser On} = 2.6 \pm 0.6$ Hz, $p = 0.014$, two-sided Wilcoxon signed-rank test) and whisking conditions ($Rate_{W, Laser Off} = 10.6 \pm 1.4$ versus $Rate_{W, Laser On} = 4.6 \pm 1.0$, $p = 0.003$, two-sided Wilcoxon signed-rank test, Fig 5B). Yet even during cortical silencing, POm activity was higher during whisking compared to quiescence ($p = 0.002$, two-sided Wilcoxon signed-rank test), suggesting a non-BC origin of state modulation. Interestingly, spike activity, especially in POm, was more phase-locked to the whisking cycle during BC silencing trials compared to control conditions (Fig 5C and 5D), suggesting that other inputs, putatively from the brainstem, carry phase information to POm. We next compared air puff responses with and without cortical silencing (active touch trial numbers during laser inactivation were not sufficient for the same analysis). BC silencing had on average no significant effect on VPM neurons' responses to air puffs (Fig 5E, 5G, and 5I). In contrast, cortical silencing abolished air puff responses in most POm neurons (8/11) and reduced rate modulation by air puffs from $0.35 \pm 0.06$ to $-0.2 \pm 0.2$ which was not significantly different from zero ($p = 0.41$).

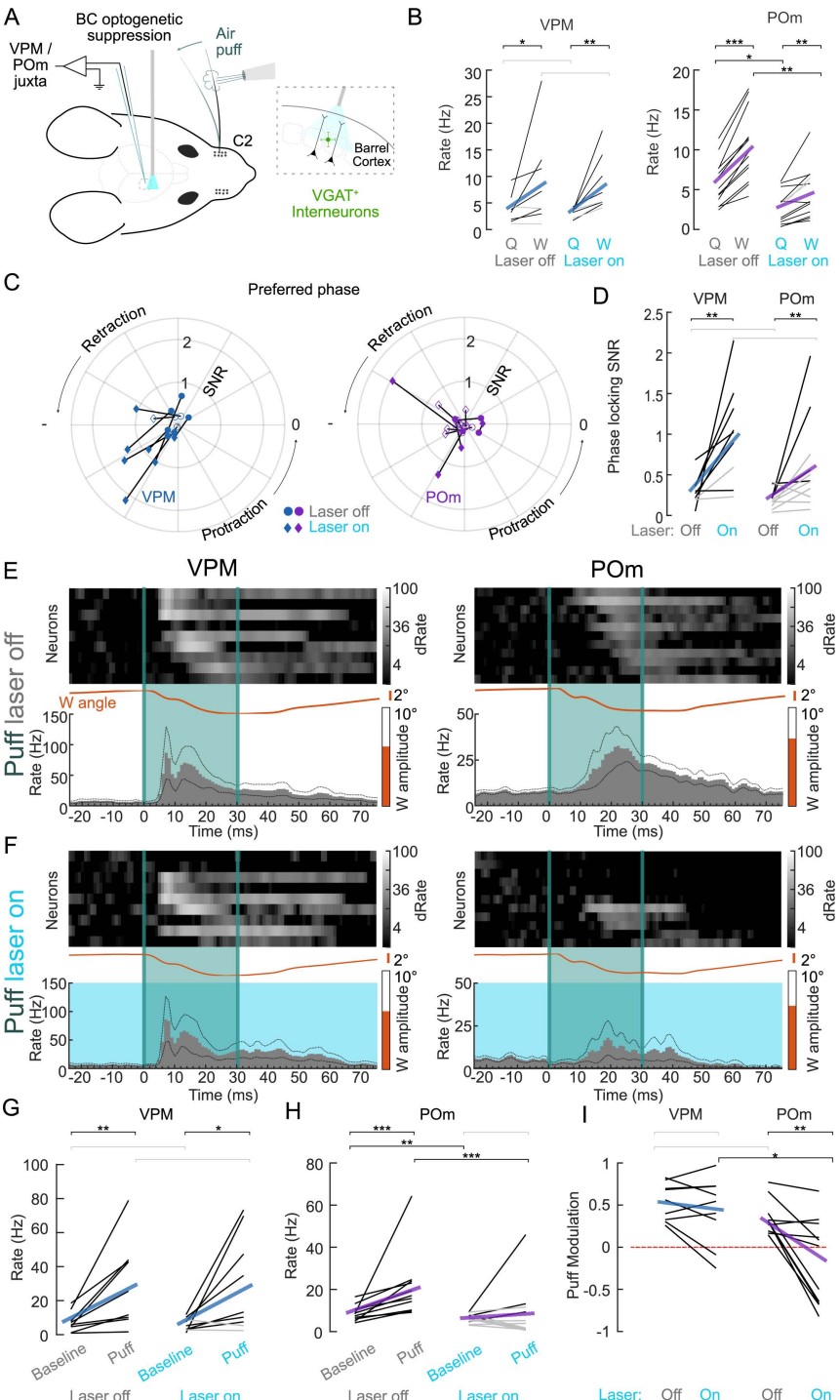

**Fig 5. Inhibition of the barrel cortex (BC) increases VPM and POm correlation with free whisking but strongly attenuates POm whisker deflection responses. (A)** Experimental setup. BC was optogenetically suppressed while juxta-cellular recordings were made from single neurons in VPM and POm. A single whisker was deflected with air puffs. **(B)** Spike rates during quiescence (Q = no whisking) and during whisking (W) during BC suppression (Laser on) and control conditions (Laser off). Individual neurons (black lines: whisking rate significantly different from base rate ($p < 0.05$, shuffle test, see Methods), gray: non-significant change) and population means (VPM blue line, POm purple line). **(C)** Polar plot of phase-SNR (radius) and preferred phase (angle) for VPM (left, blue) and POm (right, purple) neurons, during control condition (Laser Off, circles) and during BC inactivation (Laser On, diamonds). Filled markers indicate significantly phase-modulated neurons per condition. Phase zero refers to a fully protracted

position. **(D)** VPM (left) and POm (right) phase-SNRs when the laser was on or off. Individual neurons (black lines: significant phase modulation ($p < 0.05$, Kuiper test) during laser on condition, gray lines: nonsignificant phase modulation) and population means (VPM blue line, POm purple line). Percentage of neurons significantly modulated in laser on/off conditions. **(E)** VPM (left) and POm (right) responses to air puffs (teal) in control condition (Laser off). Top: heatmaps of individual neuron PSTHs normalized to baseline before air puff and ordered by peak response latency. Bottom: mean population PSTHs and whisker angles (orange). **(F)** Same neurons as in E but when the BC was inactivated (Laser on, light blue). **(G)** VPM spike rates during baseline and in response to air puffs during control conditions (Laser off) and during BC inactivation (Laser on). Baseline: 50 ms windows before, Puff: 50 ms after. Individual neurons (black lines: rate increase $p < 0.05$, gray lines: non-significant increase) and population means (blue lines). **(H)** same as G but for POm. **(I)** Rate modulation index in comparison to baseline for puff-evoked responses with and without BC inactivation. Individual neurons (black) and population means for VPM (blue) and POm (purple), zero modulation indicated by red-dashed line. Asterisks represent $p$-values (*$p < 0.05$, **$p < 0.01$, ***$p < 0.001$); 5G–5H, 5I between conditions: two-sided Wilcoxon signed-rank test; 5I between neuron populations: two-sided Wilcoxon ranked-sum test; Individual neuron comparison (5B, 5D, 5G, 5H): two-sided Wilcoxon signed-rank test; exact p values, N numbers in S1 Table. Data and code underlying this figure can be found here: https://doi.org/10.5281/zenodo.14691035. Source data for panels B, D, G, H, I in S4 Data.

## Discussion

The present study investigated the representation of active and passive whisker deflections in the somatosensory thalamus of awake mice. VPM neurons responded reliably to both active and passive whisker deflections, while POm neurons preferentially responded to passive deflections but poorly to active touches. The specific sensitivity of POm to passive deflections, which necessitates top-down cortical involvement, suggests that this nucleus may play a role in signaling behaviorally relevant events such as unexpected tactile events.

We explored several possible explanations underlying this response disparity in POm. First, tactile transmission through POm may be sensitive to whisking. Since active touches are always associated with whisking, responses to touch may be suppressed by whisking-related state modulation. Indeed, thalamic neurons showed elevated baseline firing rates during whisking, concomitantly reducing passive deflection responses, but to a lesser degree than active touch responses. This suggests that self-motion alone cannot explain the lack of POm touch responses. Moreover, this result indicates that sensory transmission in POm does not depend on whisker motor activity—a prediction arising from motor cortex—ZI interactions [11], which has not been directly tested yet. We conclude that POm differentiates between active and passive deflections and this function may have been evolutionarily optimized to be robust against changes in the whisking state.

Second, the kinematic details of active and passive deflections differed, e.g., passive deflections on average led to larger whisker curvatures and smaller angular acceleration, compared to active touches, which could lead to differences in peripheral stimulus encoding. Matching individual parameters related to impact strength could however not explain the response disparity between active and passive touches. Instead, POm neurons showed significant sensitivity to stimulus intervals, with long intervals generally causing greater responses and even significant responses to touch events when they occurred after periods of 1 s or more following the last touch (Fig 4H and 4I) or to the first touch in a touch bout (Fig 4J). Touch signals are conveyed to POm mainly from layer 5B (L5B) neurons in the BC [23–25]. While L5B neurons robustly respond to whisker touch [26,27], the action potential (AP) transfer rate across the L5B-POm synapse is limited by the short-term depression of this giant synapse [22,28]. Previous in vitro work has shown that the cutoff frequency for reliable AP transfer is around 1.7 Hz [22], which is comparable with the reliable transmission of touch events at frequencies of 1 Hz or less, estimated by the present study. This suggests that repetitive touch signals are filtered out along the strongly depressing cortico-thalamic synapse in POm and in consequence, POm is sensitive to rare events.

POm's sensitivity to rare events, on top of a de-sensitization during self-motion, may reflect POm's role in gating behaviorally relevant stimuli. In our experimental setting, head-fixed mice repeatedly touched an invariant object with the result that sensory events become tightly synchronized to self-motion. In this situation, the sensory stimulus becomes predictable, and the behavioral relevance becomes low. In contrast, during passive deflections, in which whiskers are deflected externally, sensory stimuli and self-motion are uncorrelated, thereby possibly reflecting an element of surprise. Consistent with a putative function of POm to signal unexpected events, POm was most responsive during passive deflections with long interstimulus intervals.

To a lesser extent than POm, VPM neurons were overall also more responsive to passive over active deflections. Across the VPM neuron population, we found substantial variability in deflection latencies, active touch sensitivity, and whisker phase-coding, but VPM neurons were in general more sensitive to fine kinematic, state, and self-motion parameters than POm. Confirming earlier reports [29], this encoding of fine kinematic differences suggests that VPM's role is to relay diverse high-fidelity whisker signals to the cortex.

To better understand the cortical involvement in the differential representation of active and passive touch in POm and VPM, we inhibited the BC during free whisking and passive whisker deflection trials. Inhibition of the BC strongly attenuated whisker responses in POm while increasing the whisking phase coding. This suggests that the response disparity in POm likely results from cortico-thalamic computations and that POm receives touch information from the cortex [26] and phase information from the brainstem.

A subset of POm neurons seems less dependent on BC inhibition, which could result from technical reasons of alignment and stimulation penetration. Intriguingly, the substantial response variability in POm furthermore suggests specialization within the nucleus as well as additional more complex sensory functions, especially during behavior [30–33].

In conclusion, these results suggest a robust relay of active and passive deflection via VPM, and a specific sensitivity of POm for passive deflections, which might signal unexpected events that are relayed from the BC to POm.

## Materials and methods

### Ethics statement

All experimental procedures were approved by the local governing body (Sachgebiet 54—Tierschutz, Regierung von Oberbayern, Germany, Az. 55.2.1.54-2632-73-13) and performed according to their ethical guidelines.

### Animals

Ten male wild-type C57/BL6 and 6 (4 male, 2 female) VGAT-ChR2-EYFP line 8 [34] (Jackson Labs) mice with ages between 8 and 13 weeks were used in the study.

### Habituation and head-implant surgery

For 2–4 days before surgery animals were handled by the experimenter for two sessions per day for 5–10 min until they calmly walked from hand to hand. Before the surgery, animals were anesthetized with 1% Isoflurane in $O_2$ (SurgiVet Vaporizer) and transferred to a stereotaxic device (Kopf) where the animal was given 200 mg kg$^{-1}$ metamizol s.c. for analgesia. Eyes were covered with ointment and body temperature was regulated to 37 °C. The depth of anesthesia was continuously monitored by breathing rate and lack of reflexes. The skull was exposed and the cranium was carefully cleaned and dried by scraping carefully with a surgical

knife and then covered with a thin layer of All-in-one Optibond (Kerr), cured with blue light (M+W Dental, Germany). Subsequently, an L-shaped stainless-steel head-plate was glued to the skull with Charisma (Kulzer) and dental cement (Paladur, Heraeus), covering most of the left hemisphere and the posterior third of the cranium. With Paladur, a conical bath chamber was molded around the areas above the BC and somatosensory thalamus. Finally, skin edges were glued to the cement with Vetbond (3M) and the bath chamber was sealed with Kwik-Sil (World Precision Instruments). Following the surgery, animals were given a period of at least 3 days to recover.

## Head fixation and habituation

After recovery, animals were slowly habituated to the head-fixation procedure, which did not require anesthesia. The animals were first allowed to explore the head-fixation platform and when situated appropriately, the head-plate was hooked into a custom-built aluminum fixation arm and screwed in place. The head-fixed mouse was positioned on a slightly angled stationary platform, where it could whisk freely. Animals were fixed for increasing durations (from 5 to 60 min over 3–5 sessions). In later sessions, whisker stimulation elements were added to the habituation procedure: A copper rod with 4 mm diameter, angled vertically was brought into range of the whiskers, alternating with air puffs applied to the whiskers of the mouse's left side in random intervals. White noise was played via a speaker to mask the air puff valve opening sound and room lights were switched off. Habituation proceeded until animals were spontaneously whisking but otherwise calm in the setup without excessive body movements.

## Craniotomy surgery and receptive field mapping

One day before the recording experiment, animals were anesthetized as in the previous surgery, given 200 mg kg$^{-1}$ s.c. metamizol for analgesia and head-fixed in the recording setup. The Kwik-Sil plug was removed and two small (approximately 0.5–1 mm diameter) craniotomies above somatosensory thalamus (1.6 mm lateral and 1.3 mm posterior of bregma) and BC (2.7 mm lateral and 0.75 mm posterior of bregma) were drilled while dura mater was kept intact. Craniotomies were rinsed with Ringer solution. Single units in thalamus were recorded in juxtasomal configurations as described previously [23,35]. In brief, 4.5–6 MΩ patch pipettes were pulled from borosilicate filamented glass (Hilgenberg, Germany) on a DMZ Universal puller (Zeitz Instruments, Germany). Recording and bath solutions were (mM) 135 NaCl, 5.4 KCl, 1.8 CaCl$_2$, 1 MgCl$_2$, and 5 HEPES, pH adjusted to 7.2 with NaOH. Recording solution was back-filled into the recording electrode. Signals were amplified with an ELC-01X amplifier (NPI Electronics, Germany), unfiltered and band-pass filtered signals (high pass: 300 Hz, low pass: 9,000 Hz) were digitized at 20 kHz with CED Micro 1401 mkII board and acquired using Spike2 software (both CED, UK). The recording electrode was slowly lowered into the brain and thalamic units were found by observing a large increase in electrode resistance (approximate doubling of the initial resistance) measured in voltage-clamp mode. Typically, VPM was targeted first and when a stable unit was found, its whisker responsiveness and receptive field were estimated manually by touching individual whiskers with forceps. Coordinates relative to bregma and whisker responsiveness were noted until a neuron with VPM-typical brisk responses to deflections of either C1 or C2 whiskers was found. At the end of the mapping procedure, which typically took 30–60 min, the electrode was slowly retracted and the craniotomy covered with Kwik-Sil (WPI). All but one (either C1 or C2 as determined during mapping) whiskers were trimmed approximately 5 mm from their base. Animals were brought back to their home cage. The awake recording experiments were carried out on the following day.

## Whisker stimulation

Whisker stimulation consisted of 30 ms air puffs (50 mBar) delivered via a plastic tube with a tube opening of approximately 1 mm. The opening was positioned approximately 3 cm anterior (leaving space for unimpeded whisker movements) to the stimulated whisker, which was deflected in caudal direction. The puff stimulus targeted the spared whisker approximately 2 cm radially distant from the base. The latency from command to whisker deflection was determined and corrected by the video recording for each experiment. Air puffs were applied in randomly selected five different intervals between 50 and 1,250 ms. For animals with active touch trials, the touch pole was manually moved into reach of the whisker for 2–5-min periods. The touch pole was positioned such that only large amplitude whisks (protractions of 20 ± 4°) could reach it, thus only including targeted whisking against the object (median touch rate 0.5 Hz). For each neuron, we collected an average of 257 ± 201 air puff trials and 133 ± 108 active touch events for the subset of neurons with touch trials.

## Behavior

The spared whisker was imaged from above while being backlit with a diffused infrared (850 nm) LED array with a camera (AVT Pike F-032B, Allied Vision Technologies) at 170 × 90 pixels. Video data was acquired at 625 Hz in frame-triggered mode, controlled by LabView (National Instruments), and TTL trigger pulses recorded along with the electrophysiology data (see below). An additional infrared LED with sawtooth-modulated brightness at 20 Hz was imaged for post-hoc validation of the synchronization between video files and electrophysiology. After the recording, whisker position and shape were tracked using "whisker tracker" [36] with three spline points on the whisker. Periods of grooming or when the whisker stuck to the frontal side of the touch pole were removed from the analysis. Air puff times were corrected for line delay by taking the relative onset of backward whisker deflection on the mean whisker trace as the air puff latency offset for the entire experiment (18–20 ms). The vertical axis of the touch pole was well-aligned with the camera's viewing axis, enabling a definition of the touch region as an approximately 2-pixel wide area around the profile of the pole. If the whisker touched the object, the summed brightness in the touch area decreased substantially and could be used to extract touch times. Results of this analysis were curated and corrected manually. The angular position of the whisker was normalized by subtracting the median angular position while stationary, with the convention that positive angles are protractions in the rostral direction. Whisking behavior was decomposed as described previously [14], by first band-pass filtering (zero-phase filtering with a Butterworth filter of 5th order and cutoff frequencies from 3 to 30 Hz) and then taking the Hilbert transformation to extract the instantaneous phase in MATLAB. Transitions through 0 and $\pm\pi$ of the phase signal were used to determine the extrema of the whisking cycle. Whisking amplitude and setpoint were determined, respectively, by taking the absolute difference between or mean of consecutive extrema. Periods of whisking amplitude >3° lasting more than 200 ms were defined as "whisking" and periods with amplitude <3° as "quiescence".

## Optogenetic stimulation

The stimulation of Channelrhodopsin-expressing VGAT neurons was achieved as previously [23]. In brief, 488 nm laser pulses from a solid-state laser were directed at the surface of the BC via a multimode fiber (Thorlabs, Germany; numerical aperture = 0.48, inner diameter = 125 μm) and controlled by Spike2 and CED Micro 1401 mkII board (both CED, UK). The fiber was positioned at an angle of approximately 86° (from the horizontal plane) at a distance of 2.5 mm from dura illuminating a surface area with 800 μm diameter and with an approximate

power density of 8.4 mW mm⁻². Forty Hz series of laser pulses with 50% duty cycle were used with a duration of 1 s on and 2 s off during spontaneous whisking, which reliably suppressed activity across the cortical column [28] (measured in anesthesia in a different set of experiments). For optogenetic suppression of cortical activity in the context of air puff stimuli, optogenetic 40 Hz pulses were delivered at randomized durations (0.11–2 s) and off-intervals (0.24–5 s). Air puffs were considered "Laser On puffs" if the laser was on for at least 5 ms before and 35 ms after puff onset (median laser onset 42 ms before air puff).

## Electrophysiology in awake-behaving animals

Single neurons in thalamus were recorded in juxtasomal configurations as during the mapping experiment, but adding 20 mg/mL biocytin to the recording solution. Targeting previously established VPM coordinates, when a stable neuron was found, its whisker responsiveness and receptive field were confirmed manually by touching individual whiskers with forceps. If the neuron responded to the spared whisker, the air puff protocol was run and puff responsiveness was verified by an online analysis. If confirmed, free whisking, object touch, and puff periods were alternated and repeated in ca. 2–5 min intervals. After recording from 1 to 2 neurons in VPM, POm was targeted next, where again 1–2 puff-responsive neurons were recorded. Afterward, the final neuron was filled with biocytin using current pulses [35]. Spike times were extracted based on peaks in the temporal derivative of filtered voltage traces (dV/dt) above a manually determined threshold as before [23].

## Recovery of neuron location

After filling, mice were first anesthetized with 1% Isoflurane via a face mask and subsequently euthanized with an over-dose of ketamine (200 mg kg⁻¹) and xylazine (20 mg kg⁻¹) i.p. and transcardially perfused with 4% PFA in phosphate-buffered saline (PBS). Following post-fixation in 4% PFA in PBS overnight, the brain was cut into 100 μm coronal slices and stained for cytochrome C oxidase to reveal the VPM/POm border and with an enzymatic Diaminobenzidine labeling to reveal the biocytin-filled soma and dendrite of the filled neuron as described previously [37]. Slide images with the filled soma were taken with an Olympus stereomicroscope and subsequently rigidly transformed manually to fit mouse brain atlas [38] and the locations of recorded neurons were reconstructed relative to the labeled neuron using micromanipulator positions from each recorded neuron. The average distance of labeled and unlabeled neurons included in the analysis was 210 ± 27 μm. If the reconstructed location of an unlabeled neuron was on the border of VPM and POm and thus could not unequivocally be determined, it was excluded from further analysis.

## Neural analysis

Recorded neurons were included in further analysis if they showed a significant rate increase in response to air puffs, determined by a one-sided Wilcoxon signed-rank test ($p < 0.05$, comparing 50 ms before and after air puff onset). Peri-stimulus time histograms (PSTHs) were converted to rate by dividing by the 1 ms bin size (Figs 2A, 2B, 3E, 3F, 5E, and 5F). Heatmaps of individual neuron PSTHs were normalized by dividing the raw PSTH by the average rate in the 25 ms before the deflection and the data was plotted on a log2 scale. First, spike latency was estimated by determining the relative latency of the first spike after a deflection within a 75 ms window. Modulation indices (Figs 2E, 3E, 4B, 4E, 4H, and 5I) were computed by dividing the difference of rates 50 ms before and after the deflection by their sum.

 Touch and puff responsiveness: to determine individual neuron's responsiveness to puffs and touches, each trial's spike numbers in baseline and response windows (each 50 ms) were

compared and significance was estimated with a Wilcoxon signed-rank test (one-sided for puff responses to only include neurons with stimulated principal whisker, two-sided for touch responses).

Quiescence and whisking analysis: to determine the modulation of firing rates during quiescence and whisking (Figs 3A and 5B), the whole recording was parsed into periods of whisking (amplitude > 3°) and quiescence (amplitude < 3°), as well as laser "on" or "off" for cortical silencing experiments in VGAT-Chr2 recordings. To restrict the analysis to steady states, the 50-ms period around transition times, periods of 500 ms around whisker touches, and air puffs as well as laser transitions (for laser "off" assignment) were not considered. Subsequently, spikes within the respective periods were counted and divided by the total duration. Spike rates were determined as significantly different between conditions if the rates within condition were outside of the 95% confidence interval of a 10.000-fold spike time shuffled distribution. Air puff trials were split by whisking state (Fig 3E–3G) by determining the mean whisking amplitude in the 500 ms before each air puff, with a cutoff of 3° mean amplitude between quiescent (Q) and whisking (W) puffs.

Whisking amplitudes and phase-locking: to calculate spike rates as a function of whisker amplitudes (Fig 3B), whisker amplitudes were binned by 5° intervals in which spike rates were analyzed. Significance was tested using the Matlab inbuilt significance test based on Student $T$ cumulative distribution function. Neuronal phase-locking (Figs 3C, 3D, 5C, and 5D) was determined as previously [14]. In brief, the whisker phase (amplitude > 3°) was divided into 16 bins and spikes in each bin counted relative to each bin's occurrence. Significance of tuning was determined by a circular Kuiper test [39]. Using a standard least-square regression, the resultant phase-dependent spike histogram was fit with a cosine function ($f$ = <Rate> + Amp$_{Tuning}$ * cos(phase − phase$_{Preferred}$)). A preferred phase of 0 indicates fully protracted and a phase of $\pm\pi$ indicates fully retracted positions. Phase tuning signal-to-noise ratio was computed as before [14] by SNR = [2 * Amp$_{Tuning}$/<Rate>] * $\sqrt{}$(<Rate> * $T$), with temporal window $T$ estimated from the average duration of a whisk cycle with 111 ms.

Kinematic parameter matching: to match kinematic parameters of deflections (Fig 4), mean whisker parameters (Angle, Velocity, Acceleration, Phase, Setpoint, Amplitude) of the windows 50 ms before and after each deflection, as well as each inter-deflection interval were determined and converted to absolute or relative (difference of pre and post-deflection) values. The resulting joint (puff and touch) distributions were divided into tertiles and the respective trials were grouped together. Response modulation was computed as before (50 ms pre- and post-window), but for each tertile separately and averaged across the population. Positive modulation within each condition was tested for significance by one-sided Wilcoxon signed-rank tests. To compare between touch and puff, as well as between 1st and 3rd tertile two-sided Wilcoxon signed-rank tests were used. To determine spike response probability across kinematic groups (Fig 4C, 4F, and 4I), respective PSTHs (bin size 1 ms) of individual neurons were converted to spike probability by dividing by the total number of spikes in the histogram. The mean population spiking probability was smoothed by a gaussian filter over a 10 ms window.

## Cell classification

We fed individual PSTHs to a supervised, cross-validated implementation of a logistic regression model from Scikit-Learn [40] for predicting the cells' nuclei membership to VPM or POm. Each cell's PSTH was normalized to its maximum firing rate. The dataset was split into 75% of PSTHs for training and 25% for testing. The used function was train_test_split with stratified and shuffled outputs. Most of the arguments given to the LogisticRegressionCV function were set to their default values. Only the regularization strength (*Cs*), the penalty norm (*penalty*), and the number of CPU cores deployed for cross-validation (*n_jobs*) were

set to the following values: *Cs* was an array of 50 logarithmically distributed values from 10 to the power of −3 to 2; *penalty* was set for L-2 norm; and *n_jobs* was set to −1 to use all available cores in the computer. We then assessed the significance of this prediction by permuting the correspondence between the test PSTHs and each cell with the permutation_test_score function from the Scikit-Learn model selection module. The number of permutations was set to 256 permutations and the number of CPU cores deployed was set to use all available cores. The classification was independently repeated with only the puff PSTHs, only the touch PSTHs, and both to compare their contribution to the prediction.

## Supporting information

**S1 Fig. Puff unresponsive neurons do not respond to active touch.** (**A**) Non-significant puff-responsive VPM ($n$ = 14) neuron's spike rates during baseline and after whisker deflection via Puff or Touch. Baseline: 50 ms windows before, Puff/Touch: 50 ms after deflection. Individual neurons (black: rate increase $p < 0.05$, gray: non-significant increase) and population means (blue). (**B**) Same as C but for non-significant puff responsive POm (purple, $n$ = 10). (**C**) Modulation of Puff and Touch spike rates in comparison to baseline. Individual neurons from panels a and b (black) and population means for non-significant puff responsive VPM (blue) and POm (purple), zero modulation indicated by a red dashed line. Asterisks represent $p$-values (*$p < 0.05$, **$p < 0.01$, ***$p < 0.001$); A–C between conditions: two-sided Wilcoxon signed-rank test; between neuron populations: two-sided Wilcoxon ranked-sum test. Individual neuron comparison (A, B): one-sided Wilcoxon signed-rank test; Data and code underlying this figure can be found here: https://doi.org/10.5281/zenodo.14691035. Source data in S5 Data.
(EPS)

**S2 Fig. Comparison of histological- and response-based classification of neuron identity (VPM or POm).** (Top) Neuron location in coordinate space, color-coded by the predicted membership to POm (purple circles) and VPM (blue circles) using a logistic regression model with puff and touch response PSTHs as inputs. Histologically-derived membership of the recorded neurons (ground truth) is indicated by circles for VPM and POm, respectively. (Bottom) Classifier performance for different stimulus conditions. Accuracy was determined by computing the ratio of predictions that matched the histologically derived membership of the recorded neurons (ground truth) to the total number of predictions. The $p$-value was determined by shuffling the labels of the response PSTHs to quantify the likelihood of correct predictions by chance. Data and code underlying this figure can be found here: https://doi.org/10.5281/zenodo.14691035.
(EPS)

**S3 Fig. Dependence of thalamic responses to active and passive whisker deflections on additional whisker kinematics.** (**A**) left: Distribution of angular acceleration during (±50 ms) touch (yellow) and puff (teal). Joint distribution divided into acceleration tertiles (gray-shaded bars). Right: Modulation in response to puff (teal) and touch (yellow) split by acceleration tertiles indicated left for VPM (left) and POm (right). (**B**) same as A for curvature following ([0, 50] ms) touch (yellow) and puff (teal). (**C**) same as A for change of curvature during (±50 ms) touch (yellow) and puff (teal). (**D**) same as A for velocity defined as angle change during (±50 ms) touch (yellow) and puff (teal). (**E**) same as A for absolute velocity defined as absolute value of angle change during (±50 ms) touch (yellow) and puff (teal). (**F**) same as A for setpoint just before ([−50, 0] ms) touch (yellow) and puff (teal). Note that touches by did not occur at low setpoints. (**G**) same as A for amplitude just before ([−50, 0] ms) touch

(yellow) and puff (teal). Note that touches by did not occur at low amplitudes. **(H)** same as A for angle just before ([−50, 0] ms) touch (yellow) and puff (teal). Note that touches did not occur at retracted angles. **(I)** same as A for phase just before ([−50, 0] ms) touch (yellow) and puff (teal). Asterisks represent *p*-values <0.05; two-sided Wilcoxon signed-rank test. Exact *p*-values, *N* numbers in S1 Table. Data and code underlying this figure can be found here: https://doi.org/10.5281/zenodo.14691035. Source data for bar plots in S6 Data. (EPS)

**S1 Data. Numerical data underlying Fig 2.**
(XLSX)

**S2 Data. Numerical data underlying Fig 3.**
(XLSX)

**S3 Data. Numerical data underlying Fig 4.**
(XLSX)

**S4 Data. Numerical data underlying Fig 5.**
(XLSX)

**S5 Data. Numerical data underlying S1 Fig.**
(XLSX)

**S6 Data. Numerical data underlying S3 Fig.**
(XLSX)

**S1 Table. Summary statistics for: Fig 2C, 2D, 2E; 3A, 3G, 3H, 3I; 5B, 5D, 5G, 5H and 5I.**
(EPS)

**S2 Table. Summary statistics for: Figs 4 and S3.**
(EPS)

## Acknowledgments

We thank Bert Sakmann for supporting this work, especially for fruitful discussions and for constant support in pulling electrodes. We thank Katharina Ziegler for helpful comments on the manuscript.

## Author contributions

**Conceptualization:** Anton Sumser, Alexander Groh.

**Data curation:** Anton Sumser, Rebecca Audrey Mease.

**Formal analysis:** Anton Sumser, Emilio Ulises Isaías-Camacho, Rebecca Audrey Mease.

**Funding acquisition:** Anton Sumser, Emilio Ulises Isaías-Camacho, Alexander Groh.

**Investigation:** Anton Sumser.

**Methodology:** Anton Sumser.

**Project administration:** Alexander Groh.

**Supervision:** Rebecca Audrey Mease, Alexander Groh.

**Validation:** Anton Sumser.

**Writing – original draft:** Anton Sumser, Alexander Groh.

**Writing – review & editing:** Anton Sumser, Emilio Ulises Isaías-Camacho, Rebecca Audrey Mease, Alexander Groh.

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
