## [Editor Report · Decision Letter 0]

31 Jul 2024

Dear Dr Groh, 

Thank you for submitting your manuscript entitled "Differential representation of active and passive touch in mouse somatosensory thalamus" for consideration as a Research Article by PLOS Biology.

Your manuscript has now been evaluated by the PLOS Biology editorial staff as well as by an academic editor with relevant expertise and I am writing to let you know that we would like to send your submission out for external peer review.

Once your full submission is complete, your paper will undergo a series of checks in preparation for peer review. After your manuscript has passed the checks it will be sent out for review. To provide the metadata for your submission, please Login to Editorial Manager (https://www.editorialmanager.com/pbiology) within two working days, i.e. by Aug 02 2024 11:59PM.

Kind regards,

Christian

Christian Schnell, PhD

Senior Editor

PLOS Biology

cschnell@plos.org

---

## [Decision Letter · Decision Letter 1]

4 Oct 2024

Dear Alex,

Thank you for your patience while your manuscript "Differential representation of active and passive touch in mouse somatosensory thalamus" was peer-reviewed at PLOS Biology. It has now been evaluated by the PLOS Biology editors, an Academic Editor with relevant expertise, and by several independent reviewers. 

In light of the reviews, which you will find at the end of this email, we would like to invite you to revise the work to thoroughly address the reviewers' reports.

As you will see below, the reviewers have many positive comments about your manuscript, but Reviewer 1 and Reviewer 3 in particular list a couple of concerns that need to be addressed mostly by providing additional analyses, more methodological details and further textual revisions. 

Given the extent of revision needed, we cannot make a decision about publication until we have seen the revised manuscript and your response to the reviewers' comments. Your revised manuscript is likely to be sent for further evaluation by all or a subset of the reviewers.

**IMPORTANT - SUBMITTING YOUR REVISION**

*Re-submission Checklist*

*Published Peer Review*

*PLOS Data Policy*

*Blot and Gel Data Policy*

Sincerely,

Christian

Christian Schnell, PhD

Senior Editor

PLOS Biology

cschnell@plos.org

REVIEWS:

Reviewer #1 (Scott Pluta): Sumser et al. show that neurons in the POm region of the thalamus rapidly adapt to active touch but respond reliably to passive whisker stimulation with an air puff. This challenges the previous hypothesis that the POm is more strongly driven by active touch. Some of the other results of the paper are replications of previous publications: Petty…Bruno, 2021; Moore…Kleinfeld, 2015; Ahissar, Sosnik, Haidarliu, 2000. Nonetheless, the primary focus of this paper is on passive vs. active touch, which sets it apart.

Below are a few areas of concern, which if addressed, will enhance our understanding of the results.

From figure 1, sometimes mice whisk after experiencing an air puff and sometimes they don't. Since POm spiking increases with whisking/arousal, how much of the puff response can be explained by puff-induced whisking? Please compare POm activity between puff events that did and did not elicit whisking. I realize that the presence/absence of the behavioral response could correlate with stimulus adaptation, making data selection criteria difficult. 

Since the mechanism of POm not responding to active touch is sensory adaptation, it would be useful to show this effect more explicitly, rather than having it buried in one figure panel: 4I. Perhaps it would be useful to show the POm response to the first touch in the bout of active touch, to reveal the "least adapted" response.

Does the amount of time the animal spends free-whisking before puff stimulation influence the sensory response? If so, then that would suggest the POm is updating its sensorimotor prediction via self-motion (no touch), and the puff stimulus (externally generated touch) violates that prediction.

Do you know if the POm neurons in your study are from the first (rostral) and/or second order region? Given the differences in the inputs and outputs, this could explain why some of your POm neurons have a short latency puff response, while others have a long latency response. See https://www.nature.com/articles/s41467-020-17087-7. Given that in Fig. 5F, some long latency responses persist during S1 deactivation, then we can conclude that the long latency responsive neurons are not from the second order Po?

Since VPM active touch responses are smaller than VPM puff responses at all strengths (curvature/acceleration), then is VPM also optimized for encoding passive stimulation? Or can this simply be explained by stimulus repetition rate? Figure 4H suggests that repetition rate is not the mechanism for this difference, correct? It may be useful to look at raw increases in spike rate (relative to baseline spike rate), since modulation indices get smaller as the denominator (baseline spike rate) increases.

Technical concern:

Is it possible that the air puff was hitting additional whiskers? Since POm neurons have multi-whisker receptive fields, multi-whisker stimulation could activate them better.

Reviewer #2: This article answers key questions about the long-mysterious role of PO thalamus in sensory coding of active touch. The authors successfully undertake technically challenging experiments that have not previously been possible, resulting in the resolution of many years of apparently conflicting findings in the field. Specifically, the results first replicate past findings on the role of VPM and PO in signaling vibrissa self-motion and tactile responses. They then clarify that that these sensory response properties arise from a strong dependence of PO thalamus on ongoing cortical activity in alert, actively whisking animals. This study is expertly carried-out and provides novel, important information to the field. Much of the past work on this topic has been previously published in PLOS Biology, making this a fitting journal for publishing these findings.

Reviewer #3: In this study, Sumser and colleagues investigate the potential role of first-order vs higher-order thalamic sensory relay using the rodent whisker somatosensory system as a model. This system is ideal to study active vs passive modes of touching since rodents are capable of actively moving their whiskers while they explore their environment. Previous studies have shown contrasting results concerning the responses of POm neurons to various whisker sensory stimuli.

Here, the authors perform juxtacellular recordings in awake animals and find that responses to passive deflections in POm neurons are affected by cortical feedback inactivation, indicating that they rely on the cortex to carry information about the salience of a sensory signal, while most POm neurons do not respond to active touch, when the salience might be lessened. The manuscript is well laid out, however, there is a lack of clarity in the description of some of the analysis in terms of what is included vs excluded making it hard to interpret at times and I would therefore like the authors to address the following points.

Major Points:

1) I fail to understand how such a low number of neurons from the total number of air-responsive neurons were included for further analysis, especially in the VPM (ie: 15 out of 24 VPM neurons and 11 out of 14 POm neurons), considering the low number (n=10) of minimum active touches required for neurons to be included for further analysis in Figure 2. Considering that mice move their whiskers at around 6-9Hz, it shouldn't take much more than 1 second of whisking for an object positioned in front of the animal to be touched at least ten times. Did the authors have difficulties in getting the mice to whisk? Mice can even further increase their whisking rates during active touch since the object reduced the amplitude of their whisker protractions. In effect, touches can end up being separated by only a few 10ms in previous publications (Petersen lab for example). Did the authors stipulate a minimum time separation for active touches to be included in the analysis presented in the paper so as not to overlap with each other on the PSTH of Figure 2B for example? If so, please state it in the Methods.

2) In the methods section, it is stated that "recorded neurons were only included if they showed a significant rate increase in response to air puffs".

It is therefore rather redundant for the authors to state on l.114-116 that "Both VPM and Pom neurons robustly responded to passive deflections."

Later on, the authors state that 9/15 VPM vs 4/11 POm neurons "responded" to active touch. I have not seen in the methods how this was determined in terms of statistics for this subset of cells. In any case, the final conclusion from figure 2 that "Consistent with a low sensitivity to active touch, POM's firing rate modulation was not significantly different from zero" should be attenuated since 4/11 POm actually DO respond to passive deflections. From Figure 2E, it also appears that one or more of these might even be negatively modulated by air touch. A pie chart of significantly positively or negatively responding VPM and Pom neurons might be more appropriate at this stage of the results rather than the present broad assignment of VPM = responsive to air touch, POm = not responsive to air-touch. I also believe the authors should discuss more this heterogeneity of responses of POm in the discussion. From their results, things do not appear as clear-cut as they sometimes make it out to be, especially in terms of single-cell modulation of activity vs grand average modulation.

3) Free whisking consists of two separate phases, protraction and retraction. Since air puffs are oriented to induce a backward retraction, one would expect some differences in the way neurons respond to air puffs that is dependent upon the protraction/retraction phases of whisking. Did the authors perform such analysis?

4) In figure 4, it is not clear to me whether air puffs during free whisking are included or not, since Figure 4A seems to indicate very low accelerations during air puffs compared to touches. It appears Q-puffs were included? In which case, the acceleration distributions are not comparable, since the mouse is whisking during active touch. Therefore, tertiles of theses distributions are not similar. It would be preferable to perform the analysis on the basis of the same acceleration distributions (ie: taking into account only W-puffs, and even then, those air puffs that were applied when the mouse was protracting in a similar manner do when it performs active touches). Is there are reason why this was not performed (not enough W-puffs for example)? In any case, this disparity between acceleration distributions should be further discussed.

5) Similarly, active touches are divided into tertiles in Figure 4. But if the minimum number of touches is 10, does this mean that for some cells, a "tertile" consists only of a few touches? Please elaborate on this point, since statistical significance is essential to the conclusions of Figure 4.

6) On Figure 5E-F, I find it interesting that VPM responses appear bimodal, and that the late latency component of that response is not affected by cortical inhibition despite the likelihood that this late response (because of its lateness) depends on some form of cortical feedback (from Layer 6). I would like the authors to elaborate on this. Why is this late response unaffected?

7) Similarly, for the POM, I noticed that the neurons that are mainly affected by cortical inactivation are the ones displaying a short latency response (if we look at the heatmaps, it's the ones at the top who's responses disappear, rather than the ones at the bottom of the heatmap). Again, the authors should discuss this dichotomy, possibly in the light of S1 vs S2 feedback and spread of their optogenetic light beam…

Minor points:

1) On line 82, it is mentioned that either C2 or B1 whisker is left intact, but in the methods (l. 489 and l.491), they become C2 or B2. Please correct one or the other.

2) Figure 3: in A-C, we see that about half of the "larger set" of neurons is phase-locked to free whisking. The authors then concentrate on the "air touches >10 subset" for the reminder of the Figure. Does this average of "around half the neurons" still hold for this smaller set?

---

## [Decision Letter · Decision Letter 2]

20 Feb 2025

Dear Alex,

Thank you for your patience while we considered your revised manuscript "Differential representation of active and passive touch in mouse somatosensory thalamus" for publication as a Research Article at PLOS Biology. This revised version of your manuscript has been evaluated by the PLOS Biology editors, the Academic Editor and two of the original reviewers.

Based on the reviews, we are likely to accept this manuscript for publication, provided you satisfactorily address the following data and other policy-related requests.

* We would like to suggest a different title to improve its accessibility for our broad audience and to follow journal style: 

Active and passive touch are differentially represented in the mouse somatosensory thalamus

* Please add the links to the funding agencies in the Financial Disclosure statement in the manuscript details.

* DATA POLICY:

Regardless of the method selected, please ensure that you provide the individual numerical values that underlie the summary data displayed in the following figure panels as they are essential for readers to assess your analysis and to reproduce it: 2CDE, 3AGHIJ, 4BEHJ, 5BDGHI, S1ABC and S3ABCDEFGHI.

* CODE POLICY

We expect to receive your revised manuscript within two weeks. 

*Published Peer Review History*

*Press*

Sincerely,

Christian

Christian Schnell, PhD, 

Senior Editor

cschnell@plos.org

PLOS Biology

Reviewer remarks:

Reviewer #1 (Scott Pluta): The authors have addressed all of my concerns.

Reviewer #3: The authors have performed extensive new analyses that have adequately addressed my concerns.

---

## [Editor Report · Decision Letter 3]

10 Mar 2025

Dear Alex,

Thank you for the submission of your revised Research Article "Active and passive touch are differentially represented in the mouse somatosensory thalamus" for publication in PLOS Biology. On behalf of my colleagues and the Academic Editor, Alberto Bacci, I am pleased to say that we can in principle accept your manuscript for publication, provided you address any remaining formatting and reporting issues. These will be detailed in an email you should receive within 2-3 business days from our colleagues in the journal operations team; no action is required from you until then. Please note that we will not be able to formally accept your manuscript and schedule it for publication until you have completed any requested changes.

PRESS

Sincerely, 

Christian

Christian Schnell, PhD

Senior Editor

PLOS Biology

cschnell@plos.org